# The Relationship between Sociodemographic, Professional, and Incentive Factors and Self-Reported Level of Physical Activity in the Nurse Population: A Cross-Sectional Study

**DOI:** 10.3390/ijerph19127221

**Published:** 2022-06-13

**Authors:** Katarzyna Wesołowska-Górniak, Agnieszka Nerek, Lena Serafin, Bożena Czarkowska-Pączek

**Affiliations:** Department of Clinical Nursing, Medical University of Warsaw, 02-091 Warsaw, Poland; agnieszka.nerek@wum.edu.pl (A.N.); lena.serafin@wum.edu.pl (L.S.); bozena.czarkowska-paczek@wum.edu.pl (B.C.-P.)

**Keywords:** physical activity, nurses, barriers and motivators

## Abstract

Research indicates that while nurses are aware of the benefits of physical activity (PA), their adherence to PA is low. The results of workplace interventions that increase PA are inconsistent. The study aim was identification the sociodemographic, professional, and incentive factors influencing nurses’ PA and investigation its relationship with the level of PA that they report. This study was based on observational cross-sectional research conducted among professionally active nurses working in a clinical setting (*n* = 350). The self-reported questionnaire was used to collect sociodemographic and employment data and motivators and barriers of participating in PA. The level of PA was assessed using International Physical Activity Questionnaire. The analysis revealed significant differences in the Total Physical Activity Score (TPAS) depending on the variables related to professional activity (working in a management position: *p* = 0.015; workplace: *p* = 0.01; shift type: *p* ≤ 0.002). Cluster analysis revealed that the most important statement in the group division about motivation was fear of the pain occurring after exercise. Nurses who were more motivated to be active showed a higher level of leisure-time PA than less motivated nurses. The recommendation of PA in the nursing population should be focused on increasing the leisure time PA, ensuring the appropriate time to recovery, and compliance with the principles of work ergonomics to prevent musculoskeletal disorders.

## 1. Introduction

The adherence to physical activity (PA) guidelines in connection with its efficiency in prevention of non-communicable diseases is very important. The World Health Organization (WHO) recommends an accumulation of 150 min of moderate-intensity PA or 75 min of vigorous-intensity PA each week or a combination of both for health benefits [1]. Regular PA improves adults’ overall health, quality of life, weight management, mental health, prevents depression, strengthens bones and muscles, and minimizes the risk of chronic disease and even early mortality [2,3,4]. Current research indicates that nurses’ adherence to PA is low despite their awareness of the benefits of PA and the risks associated with physical inactivity—and even though they are perceived as role models [5,6,7]. Less than quarter of the nursing population met the WHO physical activity guidelines [8,9,10]. The lack of compliance with recommendations for regular PA is observed at the very early career stages [7,11,12]. Insufficient PA in nurses is enhanced versus a global population; 27.5% of adults do not meet these guidelines (31.7% of women) [13].

Nurses’ low levels of PA place them at increased risk for chronic disease and absenteeism, which are predictors of turnover [14]. Moreover, not performing PA is associated with burnout syndrome among nurses [15]. Current reports indicate that over 30% of registered nurses are overweight or obese [5,16,17] with PA levels that are associated with Body Mass IndeX (BMI) and waist circumference [18]. Maintaining quality nursing staff is particularly important given the global nursing shortage [19]. Therefore, nurses are a target group for workplace health-promotion initiatives. The results of interventions regarding PA promotion among nurses are promising but inconsistent. Most studies assessing the impact of interventions are low to moderate in quality and should be interpreted with caution [14]. Interventions such as visual triggers, pedometers, and health coaching with texting increase PA [20], but tailored intervention programs and pedometer challenges are more effective than passive strategies such as educational material and lectures [14]. Removing barriers that discourage or prevent nurses from engaging in healthy behaviors including regular PA should be also highly recommended [5].

Prior work showed that many nurses’ barriers to PA may be associated with their job performance such as lack of time, excessive work, irregular shifts, stress, and/or exhaustion [21,22,23]. On the other hand, the nursing profession is physically demanding. Nurses can sometimes meet their PA recommendations during their shift duties [24]. The discrepancy in the results of the studies assessing nurses’ PA—not only from the methodological differences in the research, but also from the nature of the nurses’ work and the variety of activities undertaken—makes it difficult to generalize the measurements.

Therefore, the aim of this study was to identify the sociodemographic, professional, and incentive factors influencing the nurses’ PA and investigate its relationship with the level of PA they report using a self-report method. The objectives were: (1) assessment of the relationship between sociodemographic and professional variables and level of total PA and its components in nurses’ population, (2) identifying the barriers and motivators of participating in regular PA in relation to reported total PA and its components. The structure of the paper is a typical research paper structure and consists of the following sequences: Section 1, Section 2, Section 3, Section 4 and Section 5.

## 2. Materials and Methods

### 2.1. Design and Settings

The study design was based on an observational cross-sectional research protocol. The data were collected using remote data collection methods during a five-month period from March 2021 to July 2021. Online surveys are an established method in health science research, especially during the COVID-19 pandemic [25]. The online approach ensures greater completeness of the data, especially in populations larger than 300 participants [26,27]. The participants were invited to the study using a link to the survey posted on the Lime Survey platform. This link and detailed information about the study were shared across Polish nursing social networks. The information about the study was also disseminated in five hospitals and three outpatient clinics in Warsaw. The Strengthening the Reporting of Observational Studies in Epidemiology (STROBE) Statement was used to report data [28].

### 2.2. Sample

The study was conducted among professionally active Polish nurses working in a clinical setting. The sample size analysis was performed using the G * Power 3.1.9.4 software and was based on planned a priori analyses based on alpha, power and effect size for: Pearson’s correlation analysis—N = 115 (bivariate normal model, correlation); for comparisons of 3 groups based on one-way analysis of variance—N = 252 (ANOVA one-way, F test). It was assumed that the effect would be moderate, the significance level at the standard level of 0.05, and the test power at the level of 0.95, effect size = 0.3 for correlations and f = 0.25 for group comparisons. The ANOVA assumes that the compared groups are equal, and thus—about 84 people per group. 

The sample comprised 350 participants, of which 55 nurses fulfilled only sociodemographic parts of the survey; thus, these 55 were excluded from the analysis. In the analysis of PA components, data from 295 participants were included and 246 participants were analyzed for motivators and barriers of PA. We excluded all the incomplete cases. The mean imputation could have been used to deal with missing data, but it was not essential to do so because the sample was still sufficiently large even without the 55 participants.

We invited all nurses that were professionally active, and it was the only inclusion criteria. There was no age restriction due to the long period of professional activity of Polish nurses (mean age of nurses registered in Poland exceeds 53.2 years old [29]). The exclusion criterion was incomplete survey fulfillment.

### 2.3. Data Sources and Measurements

A self-reported questionnaire was used to collect the data, including sociodemographic (sex, age, place of residence, family structure) and employment data (education, clinical specialization, management position, number and type of workplace(s), total monthly workload, type of shift, and work experience). We also detailed the level of PA and its components, motivators, and barriers to participation. The level of PA was assessed using a Polish version of long form of International Physical Activity Questionnaire (IPAQ), which indicated very good repeatability (a repeatability coefficient of ρ = 0.81) and criterion validity had a median ρ of about 0.30, which indicates fair to moderate agreement between IPAQ and accelerometer measure [30]. IPAQ includes all domains that contribute to daily PA. The items were structured to provide separate domain-specific scores for walking (total walking MET), moderate-intensity (total moderate MET), and vigorous-intensity activity (total vigorous MET) within each of the work, transportation, domestic chores, gardening, and leisure-time domains. Total time engaged in walking, moderate PA, and vigorous PA and total level of weekly activity (total physical activity score- TPAS) were computed according to the guidelines as well as the category of PA (‘High’, ‘Moderate’, or ‘Low’). [31] Additionally, the data offered insights on activity related to professional work (Occupational Physical activity—OPA) and non-professional activity (Non-Occupational Physical Activity—NOPA). The NOPA was comprised of domestic chores, gardening, and leisure-time domain. 

The motivators and barriers of PA in the study population were assessed by authors’ questionnaire based on a 5-point Likert scale (1—strongly disagree; 2—rather disagree; 3—have no opinion; 4—rather agree; and 5—strongly agree). These included statements about obstacles and incentives to participate in PA. The motivators and barriers were previously identified on the basis of literature analysis [32]. This part of the questionnaire reached very good reliability (for motivators, Cronbach’s alpha reached 0.851 and for barriers 0.847). The full list of statements and their consistent validity results were included in the Appendix A
Table A1 and Table A2.

### 2.4. Statistical Methods

All data were analyzed using IBM SPSS Statistics, version 26.0. Descriptive statistics were used to assess sample characteristics. Categorical variables were described by counts and percentages; continuous variables were described by arithmetic mean (M) with standard deviation (±SD), median (Me), and interquartile range (IQR). The Kolmogorov–Smirnov test and normal plots were used to check consistency of the quantitative variable with a normal distribution. If the data was not consistent with normal distribution, then non-parametric tests were used. The Mann–Whitney U test was used to compare continuous variables between two groups. The Kruskal–Wallis H test was used to compare variables between more than two groups. The Spearman’s rank correlation analysis was used to measure the strength and direction of association between quantitative variables. A value of *p* ≤ 0.05 was considered significant.

The components share of TPAS, such as OPA and NOPA, were calculated as the ratio of professional or non-professional activity to general activity. The result was multiplied by 100%. The created index indicates the percentage share of a given type of activity in relation to the total activity. The distribution of the results of both variables was slightly deviated from the normal distribution; therefore, the analysis was performed using a parametric test (skewness values fell within the range <−2; 2>). For these indicators, a *t*-test analysis was performed for dependent samples to determine which type of activity was dominant in the tested sample.

Cronbach’s alpha coefficient was calculated to check the reliability of the part of the questionnaire devoted to motivators and barriers of participating in PA. The value of 0.7 was considered satisfactory. A two-stage cluster analysis was performed to distinguish the profiles of the respondents in terms of barriers and motivators related to taking up PA. The log-likelihood was taken as the measure of distance. The Silhouette measure reached 0.3, which proves the accuracy of the classification.

## 3. Results

### 3.1. Participants

Detailed characteristics of the study population are presented in Table 1.

### 3.2. Total Physical Activity

The average value of TPAS in the study population was 35 985 [MET minutes a week], which allows one to place this result in the category of ‘high physical activity’. The analysis revealed significant differences in the TPAS depending only on the variables related to professional activity. Nurses working in a management position displayed a lower overall level of PA than those in a non-management position (*p* = 0.015; Z = −2.42). There were significant differences in TPAS in relation to workplace (*p* = 0.01; H = 9.27). Nurses working on hospital wards showed a higher level of TPAS than nurses working as a district nurse or primary healthcare nurse (*p* = 0.041). Nurses working in rotating shifts showed a significantly higher level of TPAS than nurses working only in daytime shifts (*p* ≤ 0.002). There was no statistically significant difference in the TPAS depending on any sociodemographic variable in the study population (sex, age, place of residence, family structure) or variables related to professional activity (education, clinical specialization, working in one place or more, total monthly workload, and work experience).

### 3.3. The Occupational PA and Non-Occupational PA

The analysis showed significant differences between OPA and NOPA components of the TPAS, t (290) = 12.38; *p* < 0.001; d = 1.36; 95% CI [28.53; 39.31]. The share of OPA in TPAS (M = 58.55; ±SD = 27.27) was more than two-fold greater than that of NOPA (M = 24.63; ±SD = 22.41); see Figure 1.

The statistical analysis revealed a significant difference in OPA depending on the variables related to professional activity such as working in a management position, workplace, working in one place or more, total monthly workload, and shift type. 

Nurses working in a management position displayed a lower OPA than those in a non-management position (*p* = 0.028; Z = −2.2). Nurses working on the hospital wards showed the highest level of OPA, and their activity was higher than among nurses working in an outpatient clinic (*p* = 0.004) and as a district nurse (*p* = 0.001). Nurses working in more than one place showed higher levels of OPA than nurses who worked in one place (*p* = 0.027; Z = −2.2). A higher monthly workload implied a higher OPA (*p* = 0.001; r = 0.19) and a lower leisure-time PA (one of the components of NOPA (*p* = 0.005; r = −0.16)). Nurses working in rotating shifts showed a significantly higher level of OPA than those working in the morning shifts (*p* < 0.001) and day shifts (*p* < 0.001). The variables related to professional activity which did no differentiate the OPA were only education, having a clinical specialization, and work experience. The analysis did not show significant differences in OPA depending on any sociodemographic variable (i.e., sex, age, place of residence, and family structure).

The variables that differentiated NOPA were age, family structure, workload (but only in the leisure-time PA component), and having a clinical specialization (also only in the leisure-time PA component). The nurses characterized as being in a ‘marriage with child/children’ showed higher NOPA than single nurses or those with no children (*p* < 0.001). Older respondents had higher NOPA scores (*p* = 0.032; r = 0.13). The level of leisure-time PA was significantly higher in nurses without a clinical specialization than in nurses with a specialization (*p* = 0.019; Z = −2.35). Detailed results of the analysis are presented in the (Appendix B).

### 3.4. Motivators and Barriers to Participation in Regular PA

Two-stage cluster analysis distinguished the profiles of the respondents in terms of barriers and motivators related to taking up PA. There were two clusters in the study population: cluster 1 was motivated to PA and cluster 2 was unmotivated to PA. In cluster 1 (*n* = 137; 55.9%), we observed less agreement with the barriers than in cluster 2 (*n* = 108; 44.1%). Cluster 2 had fewer motivators (except for the statement: ‘My employer reimburses the costs of participation in sports activities, which motivates me to be active’). The most important statement in group division was ‘I am concerned about the pain that occurs after exercise, which discourages me from being active’ (importance = 1). The importance of each predictor in group classification is presented in Figure 2. The clusters did not differ in any sociodemographic variable. Education was the only variable related to professional activity that differentiated the clusters. There were more nurses with a masters of nursing in the group motivated to PA than in the unmotivated group (*p* = 0.010; χ^2^ = 6.59).

Nonsignificant differences between the clusters were noted only for two statements: ‘Due to my professional duties, I do not have time to take up physical activity’ and ‘A healthy lifestyle is currently fashionable, which encourages me to be physically active.’ These statements had a negligible share in the groups division. Detailed results of the differences between the clusters are presented in the (Appendix C). 

A comparative analysis of clusters in terms of TPAS and its components showed that nurses who are more motivated to be active show a higher level of leisure-time PA than those who are less motivated (*p* = 0.041; Z = −2.04). The TPAS and other components of PA of both groups were similar despite those differences in the motivation.

## 4. Discussion

The self-reported TPAS noted in a population of Polish nurses being professionally active is categorized as high, which is consistent with the results of surveys based on self-reported methods [33,34] and in contrast with other studies based on methods considered objective (accelerometers and pedometers), which indicated that the level of PA among nurses is generally low [5,6,7]. These analyses have contributed to testing many interventions to improve nurses’ PA, but the results are inconsistent [14]. However, some research recommends interventions to increase nurses’ PA levels [16,17]. Some of them are based on the assessment of the level of PA only in leisure time and may not reflect the actual energy expenditure during whole working day. Moreover, despite using the methods considered objective to measure total PA they may not measure PA performed during the different types of duties specific for different occupations, e.g., limitations in accurately distinguishing standing from sitting or discrepancy in being physically active or sitting. [35,36]. In connection with these observations, it seems reasonable to choose measurement methods and intervention programs according to the specification of the studied profession [37].

Of all the variables that significantly differentiate the TPAS in the present study, all belonged to the group of professional factors with no significant relationship with any sociodemographic variable. This observation led us to perform an additional analysis of the relationship between professional and sociodemographic variables and PA divided into OPA and NOPA. 

The analysis confirmed the two-fold-higher share of OPA than NOPA in TPAS, which is consistent with previous research. Most emergency nurses’ daily physical activity was accumulated at work [36]. The number of steps taken in non-working days was smaller than in working days in nurses [38,39] similar to other populations [34]. The results are consistent with observations showing that almost all of the professional variables significantly differentiated OPA (except education, having a clinical specialization, and work experience). Nurses working in rotating shifts presented a significantly higher level of general PA compared to nurses working only in daily shifts. This is consistent with results from Peplonska et al. [40]. There is also evidence that nurses working night shifts are less active than those working day shifts [41]. This could be the result of shift duties and physical engagement, e.g., nurses working night shifts were significantly less likely to perform muscle-strengthening and aerobic activity [16]. 

We also found that nurses working in management were less active than non-management nurses. In Jirathananuwat et al., the level of PA of nurse clinical practitioners and nurse managers were similar, but significant differences were seen between their OPA: The number of steps or hours during the work period was significantly greater among nurses in non-management positions [38]. The differences in OPA related to professional variables might be determined by the type of duties, including management positions, workplace or type of shift. It is emphasized by lack of differences in variables like education, having a clinical specialization, or work experience. Nurses working at the same position have similar duties during all working days, which is why the OPA level may not be modifiable and the workplace PA-increasing interventions may not be effective overall.

The lack of significant differences in TPAS depending on level of knowledge was surprising when expressed with variables such as education, having a clinical specialization, and work experience. More surprising is that the level of leisure-time PA was significantly higher in nurses not having a clinical specialization than in nurses with such as specialization. Our study confirmed previous observations: While nurses are aware of the benefits of PA and the risks associated with inactivity, they do not implement this knowledge into their own life [5,6,7]. However, it is promising that there were more nurses with a masters in nursing in the group motivated to do PA than in the unmotivated group. This suggests that knowledge may affect the willingness, but not the practice.

Our investigation revealed that a higher monthly workload led to higher OPA and lower leisure-time PA, which is partially consistent with Chappel et al. who found a positive association between the time spent engaging in moderate to vigorous PA prior to work and the time spent being sedentary during the morning shift. They suggested that for every additional minute of leisure-time PA, nurses were less active and more sedentary at work. Conversely, occupational walking time was associated with lower activity levels during leisure time [36]. 

Therefore, the quality of OPA, which depends on performed duties, should be considered to provide the most benefits. Using the potential of professional work for safe and effective implementation of activities improves nurses’ health and may change not only nurses’ well-being, but also their productivity and care quality. This has been confirmed previously [42,43,44] and is consistent with another observation resulting from the analysis of barriers and motivators of PA in the study population: The most important predictors of data clustering seen here were the statements: ‘I am concerned about the pain that occurs after exercise, which discourages me from being active’ and ‘I am concerned that physical activity will worsen my health, which discourages me from taking it’. The nursing profession has long been considered to be physically demanding [45], and perceived physical demands are associated with reported musculoskeletal disorders [46], including low back pain (LBP). Fujii et al. studied a large sample (*n* = 3066) and confirmed that in the nurses who had any type of LBP, high fear-avoidance beliefs about PA were significantly associated with experiencing chronic disabling LBP [47]. Here, the prevalence of musculoskeletal disorders was not assessed, but the experience of any musculoskeletal pain and fear of it could be an important barrier to being active. The confirmation of this conclusion could be the aim of future research. 

A surprising observation is that compliance with the statement: ‘Due to my professional duties, I do not have time to be physically active’ was not significantly different in cluster comparison analysis—this might be seen as contrasting with previous research [21,22]. However, considering the methodology of these studies, our study does not contradict them. Our observations confirmed that there was similar compliance with the statement above in the group motivated to PA and the unmotivated group, which may be related to the similarity in workload.

This study also revealed that nurses who are more motivated to be active show a higher level of PA than those who are less motivated, but only in the leisure-time PA component. The TPAS and other components of PA of both groups was similar. This observation considered the two-fold higher share of OPA in TPAS than NOPA confirmed previously. We suggest that the level of occupational physical activity is constant in nurses working at the same position, and interventions targeting an increase in PA during working hours may therefore not be effective. The inconsistency of evidence for the effectiveness of workplace health promotion programs was presented by Torquati et al., but the rationale was based on studies of limited quality and heterogeneity in outcome measures [14]. Moreover, as Chappel et al. reported, nurses are close to meeting physical activity guidelines through occupational activity alone, and workplace interventions in this population may not necessarily be needed on work days [36]. Interestingly, Henwood et al. indicated that nurses who undertook ≥30 min/day of moderate workplace activity were not healthier than those who found the same amount of physical activity in their leisure time. They concluded that activity at work fails to positively affect health and well-being [48]. These observations confirm the conclusion that intervention programs increasing physical activity in the studied population are recommended but only in leisure time, which is consistent with prior work [23]. The implications for theory and practice relate to the need to monitor the level of physical activity of nurses to develop health enhancing interventions as a permanent component of nursing management. Nurses are a role model for health behavior, therefore, taking care of proper level and quality of physical activity among nurses may not only affect their healthy and provided care, but also the actions taken by patients. Targeting activities encouraging nurses to take up physical activity should be adjusted to the work system and professional duties. Moreover, the development of nurses in the field of work ergonomics at every stage of their work also requires a lot of attention, as it can significantly improve the quality of their work activities.

### Limitations

One of the limitations of this study is the use of a self-reported questionnaire to assess the PA level in the study population. Self-reported measures tend to overestimate physical activity levels when compared with objective assessments [49]. The choice of the assessment method was targeted to assess all components of daily physical activity both during work and leisure time, which is not always possible using, for example, accelerometers or pedometers. Common tasks performed by nurses as part of their professional work including transferring patients between trolleys, beds, and chairs; repositioning patients in bed; pushing beds and wheelchairs; and carrying heavy pieces of equipment. This activity can be assessed using objective methods and may lead to contradictory results [24,50]. Heart rate monitoring should also not be considered as a direct measure of PA because heart rate can be influenced by additional stressors [24]. Multiple measures of physical activity might be more appropriate than a single self-report measure of physical activity given the multiple types of physical activity engaged in by nurses across multiple contexts.

## 5. Conclusions

The self-reported PA level in nurses that are professionally active is high, similar to results in existing work. The sociodemographic variables do not differentiate the TPAS, which may be explained by the two-fold higher share of OPA in TPAS than NOPA in our cohort. The variables that predict OPA depend only on variables that determine professional duties. The motivation to PA is related only with leisure-time PA, which explains why the results of work-place interventions regarding PA promotion are inconsistent. The most important barrier that differentiates motivation to PA participation in the study population is fear of the pain that could occur after exercise. This could be related to the common experience of musculoskeletal pain in the nurse population. Confirmation of this relationship requires more research. We recommend increasing leisure time PA, ensuring the appropriate time to recovery, and ensuring compliance with the principles of work ergonomics to prevent musculoskeletal disorders.

## Figures and Tables

**Figure 1 ijerph-19-07221-f001:**
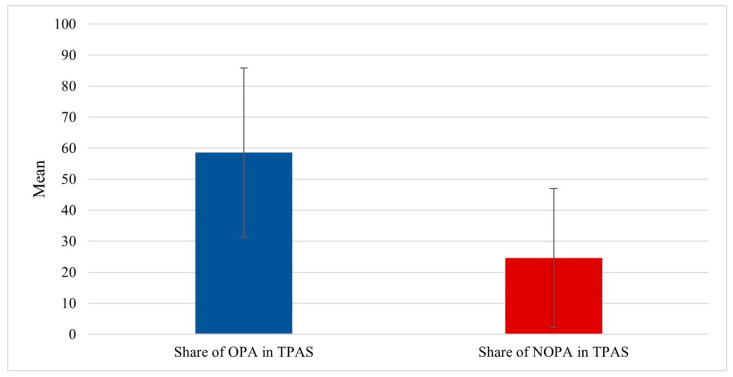
This is a figure. Average percentage of professional and non-professional activity in re-lation to total activity. OPA: Occupational Physical Activity; NOPA: Non-Occupational Physical Activity; TPAS: Total Physical Activity Score.

**Figure 2 ijerph-19-07221-f002:**
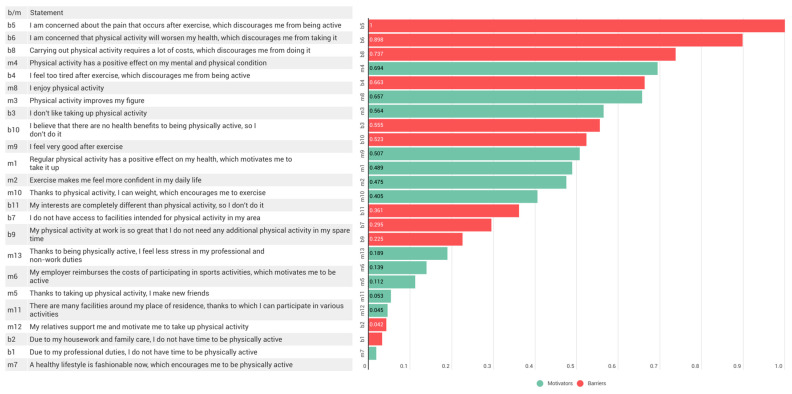
The importance of each predictor in group classification; m/b: motivator or barrier.

**Table 1 ijerph-19-07221-t001:** Detailed characteristics of the study population.

Sociodemographic Variables:	Statistics:
Sex, *n* (%)	
Woman	261 (88.5)
Man	34 (11.5)
Age [years], *M* (*±SD*)	39.48 (10.54)
Place of residence, *n* (%)	
Village	53 (18.0)
Small town	37 (12.5)
Medium-sized town	70 (23.7)
City	135 (45.8)
Family structure, *n* (%)	
Marriage and children	180 (61.0)
Single parent and child/children	33 (11.2)
Single/couple with no children	82 (27.8)
Professional activity related variables:	
Education, *n* (%)	
Bachelor of Nursing	175 (59.3)
Master of Nursing	120 (40.7)
Clinical Specialization, *n* (%)	
Yes	106 (35.9)
No	188 (63.7)
Management position, *n* (%)	
Yes	38 (12.9)
No	257 (87.1)
Workplace, *n* (%)	
Hospital Ward	231 (78.3)
Outpatient clinic	37 (12.5)
District nurse/Primary Healthcare	25 (8.5)
One place of work, *n* (%)	
Yes	156 (52.9)
No	139 (47.1)
Total monthly workload, *n* (%)	
Part-time work (less than full-time)	17 (5.8)
Full-time work	139 (47.1)
More than full-time (full-time and overtime)	139 (47.1)
Shift type, *n* (%)	
Morning shift	51 (17.3)
Daily shift	46 (15.6)
Rotating shift	195 (66.1)
Night shift	3 (1.0)
Work experience [years], *M* (±*SD*)	15.78 (11.52)

M: Mean; ±SD: Standard Deviation; Small town: <20,000 inhabitants; Medium-sized town: 20,000–100,000 inhabitants; City: >100,000 inhabitants; Morning shift: working always at the same shift from morning hours; Daily shift: working at different shifts, but always during the day; Rotating shift: working both day and night shifts; Night shift: working always the same (night) shift.

## Data Availability

The data presented in this study are available on request from the corresponding author.

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
