# Peer review of "The Relationship between Sociodemographic, Professional, and Incentive Factors and Self-Reported Level of Physical Activity in the Nurse Population: A Cross-Sectional Study"

_ijerph, 2022, doi:10.3390/ijerph19127221_

Round 1

Reviewer 1 Report

Dear author(s), thank you very much for your submission to the IJERPH and for giving me the opportunity to review your manuscript on nurses’ physical activities. Your paper is interesting and your research is worth investigation as it is astonishing that professionals who know about the importance of physical activity don’t exercise enough. Your paper shows high quality in all regards. In the following, I report on only a few issues that can help increase the quality further:

  1. Title
    1. The title should be shortened.
  2. Introduction
    1. After the statement of the research goal, you should also mention the methodology you’re using.
    2. At the end of the introduction, you should explain the paper’s structure.
  3. Methodology
    1. Please state when the data collection took place.
  4. Results
    1. Figure 2 should be rotated by 90 degrees so that it fits on a page completely.
  5. Discussion
    1. Please discuss the theoretical and practical implications.
    2. Pleas outline future research beyond your study’s limitations.
  6. References
    1. Please check for additional recent (last 2-3 years) relevant articles in IJERPH.
  7. Language
    1. While the language level is high in general, the text still needs a proofreading. Several times, the article before a noun is missing.
    2. No colon after sub-headlines.

I hope you find my comments helpful. Good luck with your revision!

Author Response

Reviewer 1:

  1. Title
    1. The title should be shortened.

Authors’ response: The title has been corrected according to the comment (‘The relationship between sociodemographic, professional, and incentive factors and self-reported level of physical activity in nurses’ population.  A cross-sectional study’)

  1. Introduction
    1. After the statement of the research goal, you should also mention the methodology you’re using.
    2. At the end of the introduction, you should explain the paper’s structure.

Authors’ response: The ‘Introduction’ section  has been corrected according to the comments

  1. Methodology
    1. Please state when the data collection took place.

Authors’ response: The details of data collection period  have been added according to the comment

  1. Results
    1. Figure 2 should be rotated by 90 degrees so that it fits on a page completely.

Authors’ response: Figure 2 was added to the manuscript only to indicate its place in the text. The picture with full quality  has been added separately during the submission process. We will add the rotated version to compare and to decide which one better fits to the page.

  1. Discussion
    1. Please discuss the theoretical and practical implications.
    2. Pleas outline future research beyond your study’s limitations.

Authors’ response: the theoretical and practical implications as well as determining the direction of future research have been added to the Discussion section.

  1. References
    1. Please check for additional recent (last 2-3 years) relevant articles in IJERPH.

Authors’ response: The recent articles has been checked and added with accordance to the main content of the manuscript

  1. Language
    1. While the language level is high in general, the text still needs a proofreading. Several times, the article before a noun is missing.

Authors’ response: The manuscript was edited by the expert staff of American Manuscript Editors. We will add the certificate to the supplemental materials and we will check whole manuscript to verify missing articles.

  1. No colon after sub-headlines.

Authors’ response: The colons  has been corrected according to the comment

Reviewer 2 Report

Katarzyna Wesolowska-Gorniak et al presented a manuscript titled: “The sociodemographic, professional, and incentive factors influencing the nurses’ physical activity and its relationship with the level of physical activity they report: A cross-sectional study”. While the study shows some interesting results, there are some ambiguities and flaws which need to be revised.

1.     There are more than few grammatical errors and typos. Moreover, some sentences have a “weird” construction. Please thoroughly check and revise the whole manuscript.

2.     Please add the statistical values to the Abstract when referring to your results.

3.     You should state the exact period (which months and during which year) during which you conducted the data collection.

4.     Please provide more information regarding the sample size analyses. Did you conduct the analysis for the population (total number of nurses in Poland) or statistical power? Please elaborate this. If you conducted sample analysis regarding statistical power, you should state which parameter you used and provide the exact values of each group.

5.     Were there any other inclusion/exclusion criteria besides the incomplete survey fulfillment? Please add this.

6.     Range (min-max) is not an appropriate presentation method of the variables for this type of a study. Please reconduct your analyses and provide the IQR instead of the min-max.

7.     Table 1. – I really think that stating: “This is a Table” in the caption is unnecessary. Remove that. Furthermore, do not state “study population” but “study sample”. Revise the whole manuscript regarding this.

8.     Please add the symbol ± between the mean and SD. Revise the whole manuscript.

9.     Also, add the unit of the quantitative variable presented in Tables/Text.

10.  When using any abbreviation for the first time, please provide the full name of the abbreviated term.

Reviewer 3 Report

In this manuscript, a cross-sectional study is presented in which predictors of nurses’ physical activity were examined. The topic falls squarely within the domain covered by this journal. The appropriate sample size and the high quality of the writing are desirable features of the manuscript. These positive impressions notwithstanding, I have several concerns about the manuscript in its current form:

1.         On l. 55, the word “the” in the phrase “of which the 55” can be deleted.

2.         With regard to the sentence on l. 89-91, mean imputation could have been used to deal with missing data, but it wasn’t essential to do so because the sample was still sufficiently large even without the 55 participants.

3.         Given the noted overestimation of physical activity using self-report measures, it would be helpful to describe the criterion-related validity of the IPAQ (especially in terms of how strongly it correlates with objective assessments of physical activity).

4.         On l. 114-115, it should say “assessed by the authors’ questionnaire.”

5.         On l. 128, it should be “with a normal distribution.”

6.         With regard to the finding presented on l. 234-236, could having a master’s degree in nursing have been a proxy for (monetary) income?

7.         On l. 242, it should be “Nonsignificant differences….”

8.         On l. 256, it should be “These analyses have contributed….”

9.         On l. 267, it should be “in the present study.”

10.       On l. 292, it should be “working days, which is why.”

11.       On l. 293, it should be “effective in overly.”

12.       On l. 324-325, it should be “prevalence of musculoskeletal disorders was not assessed.”

13.       With regard to the sentence on l. 365-367, multiple measures of physical activity might be more appropriate than a single self-report measure of physical activity given the multiple types of physical activity engaged in by nurses across multiple contexts.

14.       On l. 372, it should be “variables that predict OPA.”

15.       In the References section, closer attention should be paid to adhering more consistently to the IJERPH format.
